# International Clones of High Risk of Acinetobacter Baumannii—Definitions, History, Properties and Perspectives

**DOI:** 10.3390/microorganisms11082115

**Published:** 2023-08-19

**Authors:** Andrey Shelenkov, Vasiliy Akimkin, Yulia Mikhaylova

**Affiliations:** Central Research Institute of Epidemiology, Novogireevskaya Str., 3a, 111123 Moscow, Russia

**Keywords:** *Acinetobacter baumannii*, international clones of high risk, global clones, genomic epidemiology, multidrug resistance, IC10, OXA-51-like, cgMLST profiles

## Abstract

*Acinetobacter baumannii* is a Gram-negative coccobacillus with exceptional survival skills in an unfavorable environment and the ability to rapidly acquire antibiotic resistance, making it one of the most successful hospital pathogens worldwide, representing a serious threat to public health. The global dissemination of *A. baumannii* is driven by several lineages named ‘international clones of high risk’ (ICs), two of which were first revealed in the 1970s. Epidemiological surveillance is a crucial tool for controlling the spread of this pathogen, which currently increasingly involves whole genome sequencing. However, the assignment of a particular *A. baumannii* isolate to some IC based on its genomic sequence is not always straightforward and requires some computational skills from researchers, while the definitions found in the literature are sometimes controversial. In this review, we will focus on *A. baumannii* typing tools suitable for IC determination, provide data to easily determine IC assignment based on MLST sequence type (ST) and intrinsic *bla_OXA-51_*_-like_ gene variants, discuss the history and current spread data of nine known ICs, IC1-IC9, and investigate the representation of ICs in public databases. MLST and cgMLST profiles, as well as OXA-51-like presence data are provided for all isolates available in GenBank. The possible emergence of a novel *A. baumannii* international clone, IC10, will be discussed.

## 1. Introduction

*Acinetobacter* spp. are glucose-non-fermentative, non-motile aerobic Gram-negative coccobacilli [1] forming clusters of closely related species, which were traditionally considered as a complex, including *Acinetobacter calcoaceticus*, *Acinetobacter baumannii*, *Acinetobacter nosocomialis*, and *Acinetobacter pittii* [2], together with several other members proposed for inclusion later [3]. The species from the complex are closely related genetically and exhibit similar phenotypic and biochemical properties, so molecular or genomic methods are required to perform their accurate identification [4]. *A. baumannii*, named in honor of the American bacteriologists Paul and Linda Baumann, is an opportunistic human pathogen, the unique persistence properties of which have made it one of the most successful bacterial species responsible for healthcare-associated infections (HAI) worldwide, particularly in intensive care units (ICU) [5].

In 1971, when the *Acinetobacter* genus was officially recognized, clinical *A. baumannii* isolates were usually susceptible to commonly used antimicrobial drugs, including ampicillin, gentamicin, and chloramphenicol. However, within a few years, it managed to acquire various resistance determinants and became a significant nosocomial pathogen in the late 1970s [6]. It was at this time that two *A. baumannii* clones, later defined as global clones 1 and 2, emerged and started their five-decade history of global dissemination and acquisition of multi-, extensive-, and pan-drug resistance to antibiotics [7]. *A. baumannii* can cause infections of the skin and soft tissue, urinary tract infections, meningitis, bacteremia, and pneumonia, the latter being the most frequently reported [8]. Currently, clinical *A. baumannii* isolates usually show resistance to all first-line antibiotics, and polymyxins, tigecycline, and ampicillin/sulbactam have become the last resort of treatment for extensively drug-resistant (XDR) isolates in many cases [9,10,11]. In 2017, carbapenem-resistant *A. baumannii* was assigned a critical priority level by the World Health Organization (WHO) in terms of developing novel control, treatment, and prevention measures [12].

In order to cope with the threat imposed by this pathogen, healthcare professionals and researchers should “know the enemy”, i.e., possess the tools to effectively perform epidemiological surveillance, investigate the spread of high-risk strains, clones, or lineages, and elucidate the mechanisms of antibiotic resistance acquisition and transfer. One of the most widely used classification schemes for *A. baumannii* and several other pathogens is based on the concept of “international clones”, which was initially defined in 2010 [13] as at least two isolates that were collected from at least two countries showing a high degree of clonality [14]. The typing techniques used to identify whether a particular isolate belongs to a known international clone were developed in step with the evolution of microbiological, molecular, and genomic methods of bacterial analysis.

In this review, we will describe the definitions of international clones (ICs) for *A. baumannii*, elucidate the methods used for their classification, describe the properties and characteristics of particular ICs, investigate the representation of the isolates belonging to known IC1-IC9 in public databases, and track their spread across the world. In particular, comprehensive typing data will be provided for all the isolates deposited in GenBank. The possible emergence of novel ICs and future perspectives on their development are discussed.

## 2. Isolate Typing Methods for *A. baumannii*

*A. baumannii* exhibits an outstanding tendency to cause hospital outbreaks [15,16], which became an additional problem during the COVID-19 pandemic [17]. One of the reasons for this lies in its exceptional ability to survive for up to 60 days on various surfaces, including human skin and dry inanimate materials such as filter paper, glass, cotton, and metal, as well as on hospital surfaces [18,19], which underlies the necessity for accurate and reliable isolate typing in order to discriminate outbreak-causing strains. In general, typing techniques can be divided into phenotypic-based methods, DNA banding pattern-based methods, partial sequence-based methods, and whole genome sequencing (WGS)-based methods [16]. A brief description of these methods, together with their pros and cons, is presented in Table 1. Here, only a brief description of the typing methods suitable for distinguishing international clones is presented. A detailed analysis and comparison of these methods can be found elsewhere [16,20,21].

Although matrix-assisted laser desorption ionization-time of flight (MALDI-TOF) technology is widely used for rapid bacterial species identification in clinical settings [32], the insufficiency of such an approach in discriminating outbreak strains of *A. baumannii* was noticed previously [33].

Ribotyping and amplified fragment length polymorphism (AFLP) were the first approaches that demonstrated the clonal nature of *A. baumannii* by defining two global clones (GC) in 1996 [34], which were first named ‘European clones‘ and now are designated as GC1 and GC2, or IC1 and IC2. The AFLP fingerprints consisted of 50 well-separated bands, and a delineation level of 89% was chosen by the authors to reveal two clusters of the outbreak strains. Currently, these approaches are not widely used for identification purposes due to their high cost and low inter-reproducibility, and pulsed-field gel electrophoresis (PFGE) possessing higher discriminatory power is more popular [20]. PFGE was previously considered a gold standard for bacterial typing for a limited number of isolates and still represents a useful complementary epidemiological tool at a large scale [20,21]. In this approach, the genome under investigation is cut by a rare-cutting restriction enzyme, producing several relatively large fragments that are separated using electrophoresis, periodically changing the direction of an electric field and enabling the resolution of large fragments of up to one Mbp. The resulting 20-band patterns are analyzed, usually using computer-assisted techniques, and the common similarity cutoff is 80–85% [35].

Another powerful typing approach is repetitive sequencing-based (rep)-PCR. It examines the distance of DNA fragments located between repetitive sequences dispersed in bacterial genomes using the primers designed to anneal under relatively stringent amplification conditions on these repetitions [21]. The rep-PCR technique is rapid, flexible, and accessible, but it suffers from problems with banding pattern interpretation due to scarce biological explanation, while another problem is inter-laboratory reproducibility [36]. However, semi-automated (rep)-PCR marketed under the name of DiversiLab™ (bioMérieux, Marcy l’Etoile, France) overcame these limitations, and eight international clones (WW1-WW8, now called IC1-IC8) were first identified using this very technique [23]. An example of rep-PCR patterns is shown in Figure 1.

Typing methods based on partial sequencing constitute another group of techniques available for *A. baumannii*. One of them found to be particularly useful for IC typing was *bla_OXA-51_*_-like_ gene sequencing. *A. baumannii* has a naturally occurring class D carbapenemase gene intrinsic to this species, which is rarely present in other *Acinetobacter* [37,38], and the first report of this gene described *bla_OXA-51_* [39]. Since then, a large number of closely related variants including OXA numbers 64, 65, 66, 67, 68, 69, 70, 71, 75, 92, 94, 95, etc., were found, which were collectively referred to as ‘*bla_OXA-51_*_-like_’ genes, and the presence of one such variant in *A. baumannii* isolates analyzed was confirmed both by PCR [40,41] and WGS [42]. This typing approach remains useful for various purposes due to its low cost and rather high discriminatory ability but should be used in combination with some other techniques [42,43].

Currently, the honorary title of the gold standard is shifting towards multilocus sequence typing (MLST), which was first proposed in 1998 for *Neisseria meningitidis* [44]. This technique represents an unambiguous procedure for characterizing isolates of bacterial species using the sequences of 400–500 bp internal fragments of (usually) seven housekeeping genes. For each gene, the different sequences present within a particular species are assigned as distinct alleles, and the combination of alleles at each of the seven loci constitutes a sequence type (ST) [45]. The sequences of the corresponding alleles for a particular isolate can be obtained either by polymerase chain reaction (PCR) with a standard set of primers or by WGS and subsequent database search. The sequences, alleles, and sequence types are stored in dedicated publicly available databases, the most widely used of which is PubMLST.org [46]. Two different typing schemes known as ‘Pasteur’ [13] and ‘Oxford’ [26], each having its own advantages and disadvantages, were proposed for *A. baumannii*. The former scheme includes *cpn60*, *fusA*, *gltA*, *pyrG*, *recA*, *rplB,* and *rpoB* genes, whereas the latter relies on *gltA*, *gyrB*, *gdhB*, *recA*, *cpn60*, *gpi,* and *rpoD*. Although they have three common genes, the Pasteur scheme appears to be less discriminant for closely related isolates, but less affected by homologous recombination and more appropriate for precise classification within clonal groups [30], whereas the Oxford scheme possesses higher discriminatory power, but suffers from problems due to recombination, *gdhB* gene paralogy, and technical artifacts [47]. The number of available STs in the PubMLST database is 2850 for Oxford profiles and 2262 for Pasteur profiles (25 March 2023). An example of an ST is shown in Table 2.

Other features of *A. baumannii* that can potentially be used for isolate typing and classification purposes include capsule synthesis loci (K, or KL-loci) and lipooligosaccharide outer core loci (OCL) [28,48]. Capsular polysaccharide (CPS) is a critical determinant of bacterial virulence and phage susceptibility, which makes it a valuable epidemiological marker, and OCL also exhibits a variation in *A. baumannii* [28]. CPS gene cluster consists of about 30 genes, whereas the OC locus includes just five, and each distinct gene cluster found between the flanking genes is assigned a unique number identifying the locus type. Similar to MLST, the gene sequences and corresponding KL and OCL types are stored in a public database [49], and typing procedures can be performed using dedicated software online or offline [50]. However, due to the limited number of KL and OCL types for *A. baumannii* (237 and 22, respectively, on 20 March 2023) in comparison to, e.g., available MLST profiles, this classification can provide an added epidemiological value only in combination with other schemes like MLST [51].

Despite the wide use of MLST, the rapidly increasing availability of WGS and the number of publicly available *A. baumannii* genomes suggests a transition towards a more comprehensive typing scheme that can take full advantage of the full genome sequence availability. The technique proposing such a scheme is called core-genome multilocus sequence typing (cgMLST). It was first defined for *A. baumannii* in 2017 [29] and comprised 2390 genes that can be successfully found in most (up to 99%) existing isolates. Allele calling and result comparison is performed using dedicated software (e.g., [52]) and a publicly available database (https://www.cgmlst.org/ncs/schema/schema/3956907/ for *A. baumannii* [29], accessed on 21 May 2023). Currently, cgMLST is considered the most suitable and comprehensive method for genotyping *A. baumannii* in epidemiological research [20,53], but its use is limited by the availability of sequencing equipment in microbiology laboratories and hospitals, as well as by the high cost of WGS in comparison to traditional techniques for a relatively low number of isolates.

In conclusion, WGS, MLST, DiversiLab, and *bla_OXA-51_*_-like_ gene typing can be useful for large-scale and population studies. WGS and PFGE can be applied for local short-term outbreak investigations, and only WGS can be applied to fine-scale typing, including evolutionary genomic studies and transmission route detection.

## 3. International Clone (IC) Definitions, Attributes and Properties

In this section, we describe the diversity of the possible definitions of international clones, which are closely related to the methods used for isolate typing. Currently, various databases and software were developed for typing *A. baumannii* isolates using MLST, cgMLST, KL, and OCL based on submitted WGS data. However, to the best of our knowledge, no software has been developed for the direct assignment of a particular isolate to a known IC. In various reports, the authors usually use a detected ST to deduce the possible IC for their isolates [54,55,56,57], which is straightforward when the isolate belongs to a well-known ST already assigned to IC (e.g., ST1 and ST2 in the Pasteur scheme belong to IC1 and IC2, respectively). However, what if the ST detected for the isolate is rather rare or even novel? The provision of a solution for this problem for researchers who do not have elaborated computational tools at hand was the main motivation for writing this review. In order to better understand ICs, let us go back in history.

Although the terms ‘European clones’ and later, ‘global clones’ were known since the 1990s when the clonal nature of *A. baumannii* was demonstrated [34], the concept of IC was first introduced in 2010 in the work of Diancourt et al. [13], where Pasteur MLST scheme was defined and compared with AFLP typing. In this paper, 59 STs were defined and grouped by minimum spanning tree analysis into clusters named clonal complexes (CC), where MLST profiles belonging to the same CC had a single allelic mismatch with at least one other member of the group. Three of these CCs corresponded to IC1 (ST1 was the central profile of the group), IC2 (ST2), and IC3 (ST3), and a cluster named ST15 was revealed, which corresponded to IC4 not defined at that time. In the same year, eight worldwide (WW) clonal lineages, which later became ICs, were defined by molecular typing using rep-PCR [23]. However, the association between ICs and STs, which could make the assignment easier, was established by computational, rather than molecular, methods, with the introduction of the eBURST algorithm, which divides an MLST data set of any size into groups of related isolates and CCs and predicts the founding (ancestral) genotype of each CC [58]. This algorithm was widely used for defining CCs of various bacterial species [59,60], and it was later applied to associate *A. baumannii* ICs with the corresponding dominant STs from both the Pasteur and Oxford schemes developed [61].

It should be noted that CC is a computationally derived concept, which indicates the similarity of MLST-based profiles for the isolates, and not all CCs can be assigned to a particular IC; although CC and IC have a good correlation, the latter is a broader notion that better reflects the epidemiological characteristics of the clones.

An example of using eBURST is shown in Figure 2.

Similar to any other mathematical model of a biological process, the eBURST classification does not provide a univocal grouping of STs into ICs. The main problem here is that new STs are added to public databases rather frequently when researchers report the isolates possessing such properties. Initially, the Pasteur MLST scheme included 59 STs in 2010 [13], and now it includes more than 2000 STs (https://pubmlst.org/bigsdb?db=pubmlst_abaumannii_seqdef&set_id=2&page=downloadProfiles&scheme_id=2, accessed on 28 March 2023), which makes the determination of CC more complicated and could produce slightly different results. This issue is further discussed in Section 6.

The complete eBURST results for the currently available Pasteur ST are presented in Appendix A.

Almost concurrently with the development of IC typing approaches mentioned above, *bla_OXA-51_*_-like_ gene variants were introduced for this purpose. First, a study investigating the association between *bla_OXA-51_* variants and DiversiLab Rep-PCR-based typing for *A. baumannii* isolates was published in 2012 [62]. The correlation between particular variants and IC was high, but not perfect. In another more recent study, it was revealed that some isolates belonging to a particular IC possessed *bla_OXA-51_*_-like_ genes that differed from the commonly found variant for this group, and some variants associated with a particular IC were revealed in the isolate that obviously did not belong to that IC [63]. In yet another study, the correct typing for IC1-4, 7, and 8 by *bla_OXA-51_*_-like_ gene typing was achieved and performed no worse than MLST [27]. However, a recent investigation suggests re-evaluating the correlation between MLST and *bla_OXA-51_*_-like_ gene typing because of some contradictory results obtained [63].

In conclusion, eight ICs were first defined by the similarity of rep-PCR band pattern profiles, whereas IC1 and IC2 were revealed earlier by AFLP. PFGE patterns were also applied, but their use for IC definition was limited since this method was considered unsuitable for international population studies [27]. Then, an MLST-based approach was proposed, as well as *bla_OXA-51_*_-like_ gene typing, which showed a good correlation with previously developed techniques. Later, the cgMLST-based approaches were introduced. However, even the latter has some issues like choosing appropriate isolates for reference profiles, as well as a smaller number of cgMLST allele differences for isolates belonging to different MLST-based STs than for the isolates belonging to the same ST in some cases [64].

The attributes of the nine currently recognized ICs, by which their typing is usually performed, are presented in Table 3. However, exclusions and additions can occur. Nevertheless, if a particular isolate was found to possess the ‘founder’ ST and most common variant of the *bla_OXA-51_*_-like_ gene, it can be assigned to a corresponding IC with a very high degree of reliability. In Table 3, ICs are numbered according to the definitions provided by PasteurMLST and DiversiLab Rep-PCR-based typing. The data previously provided in the literature [13,27,30] were used in the table.

The maximum likelihood phylogenetic tree for the amino acid sequences of the OXA-51 variants given in Table 3 is shown in Appendix A. The bootstrap consensus tree was inferred from 1000 replicates using MEGA X [66]. The variants belonging to the same IC are mostly clustered together. The tree also conforms well to previous nucleotide-based studies of *bla_OXA-51_* variants [67].

Currently, all these approaches are used under various conditions, and the choice is mostly guided by the availability of corresponding equipment and output effectiveness in terms of cost, time, and labor.

Meanwhile, the combination of MLST, *bla_OXA-51_*_-like_*,* and possibly, cgMLST typing is preferred when whole genome sequences are available for the isolates under investigation. Such a combination allowed the revealing of the ‘youngest’ IC, namely, IC9 (CC464), which was introduced in 2019 [68], although CC85 (or CC6), which became a part of IC9, was previously considered as an emerging clone [41].

The readers who have read this far are probably confused with all these partially overlapping, yet not perfect, definitions of ICs and approaches to their typing. In order to overcome this confusion, we have analyzed more than 17,000 complete and partial *A. baumannii* genomes available in Genbank (https://www.ncbi.nlm.nih.gov/genbank/, accessed on 12 January 2023) to detect their MLST-based sequence types, carriage of OXA-51 variants, and cgMLST profiles. We also analyzed the literature data associating particular STs and variants of *A. baumannii* isolates with nine known ICs using experimental approaches. Using all of the above, we deduced a reliable association between less common STs and OXA-51 variants and ICs, the results of which are shown in Table 4. We used eBURST analysis data for single-locus variants (i.e., STs should differ by no more than one MLST allele from the central ST for a particular group), verified *bla_OXA-51_* variants using the beta-lactamase database BLDB [69], and used experimental data available in the literature (in particular, for ST6, ST85, ST460 [41]; ST81 [70]; ST82 (OXA-128) [27]; ST156 [63]; ST175, ST945, ST986 [71]; ST730 [55]; ST1106 (OXA-371) [42]; ST1196 (OXA-65) [72]). We also compared the cgMLST profiles of the representative isolates. Some OXA-51 variants or STs were possibly not revealed due to the absence of the genomes representing them in public databases.

## 4. Current Situation and Historical Spread of IC1-IC9 Worldwide

The clonal population structure of *A. baumannii* was confirmed by comparative MLST and other means of typing of different strains that were isolated from hospitalized patients in multiple countries and continents. The results demonstrated the occurrence of at least nine successful clones of *A. baumannii* dominated by the first three clones 1–3 [13,23,73,74,75]. In most cases, multidrug- and sometimes pan-drug-resistance are the main features of *A. baumannii* isolates belonging to any IC [23,64,74]. This fact clearly reflects the antibiotic pressure in the hospital environment. Although there exists a shared tendency for antimicrobial resistance, there are some significant differences between the ICs. However, IC2 seems to be the most widespread clonal lineage and is frequently associated with carbapenem resistance [23,76]. ICs 1 and 2 show extensive global dissemination in more than 30 countries, supplemented by a significant repertoire of antimicrobial resistance determinants [7].

The current reports on all ICs are described below with a focus on the less studied clones IC4-IC9. The comprehensive descriptions of IC1 can be found in [7], and IC2 was the focus of many investigations (see above).

The main point here is that the global spread of ICs demonstrates their cosmopolitan character. Although some previous studies considered various clones like IC4, IC6, and IC7 to be bound, or even endemic, to some countries or continents [54,55,77,78], recent studies showed that *A. baumannii* isolates belonging to up to four or five ICs could be simultaneously revealed in a single hospital during limited periods of time [74,79]. Thus, more appropriate statements regarding the *A. baumannii* spread should include the prevalence of particular ICs in a given region during some period.

### 4.1. IC1, IC2 and IC3

Members of the three major ‘European’ lineages are associated with the majority of hospital outbreaks caused by *A. baumannii* worldwide and a higher percentage of resistance to many antimicrobial drugs [17,19,80,81]. IC2 isolates frequently showed higher resistance rates to all antimicrobial agents than the other genotypes [82]. The major concern is the acquisition of carbapenem resistance, which is usually mediated by carbapenem-hydrolyzing class D β-lactamases OXA-23, OXA-24/40, and OXA-58 in *A. baumannii* [83]. Chromosomally encoded OXA-51-like carbapenemases are usually expressed at a low level and do not confer resistance by themselves, but the transposition of ISAba1 or ISAba9 promoters upstream of the *bla_OXA-51_*_-like_ gene can cause its overexpression, which mostly occurs in the absence of other carbapenemases [9].

The analysis of the genomes of 45 *A. baumannii* isolates belonging to IC1 during the 30-year period allowed the detection of many recombination events within IC1 genomes, by means of which the nucleotide diversity >20 times the substitution rate was introduced into the population [7]. These events were distributed non-randomly within the genomes and produced notable diversity within the loci encoding the capsular polysaccharide, the outer core lipooligosaccharide, and the outer membrane protein CarO. The spreading of mutations conferring antimicrobial resistance in the *gyrA* and *parC* genes and insertion sequences activating the *ampC* gene were also observed. IC1 isolates accumulated resistance to novel anibiotics by means of a plethora of genetic mechanisms, including the acquisition of plasmids and transposons or mutations in chromosomal genes [84]. Holt et al. showed that IC1-isolates have diversified into multiple successful extensively antibiotic-resistant subclones which differ in their surface structures [7].

Besides being considered one of the major clones in the past, IC3 in the last decade seems to be less prevalent, with a rather low number of reports worldwide, which were associated with non-human origin [67,85]. However, the emergence of carbapenem-resistant clinical IC3 isolates in Peru was also reported in 2018 [86].

The intra-clonal diversity of phenotypic and genotypic resistance characteristics among isolates of IC2 can be attributed to the scattered spread of these clones worldwide, which resulted in access to a range of varied pools of transmissible resistance elements [87]. The more recent emergence of IC3 was also proposed because of the lower intra-clonal diversity within this lineage compared to IC2, and it was suggested that IC1–IC3 most likely played a supplementary role in the global dissemination of *A. baumannii* infections [13].

IC1 and IC2 were identified in all continents, confirming their ubiquitous spread and affirming their attribution to global clones. IC3 seems to not play a major role in recent years with sporadic isolates revealed in Spain, the USA, and South Africa.

### 4.2. IC4 and IC5

In Brazil, as well as Argentina, Chile, and Paraguay, the major carbapenemase-producing clones were found to belong to clonal complexes (CCs) 15 and 79, which corresponded to IC4 and IC5, respectively [88,89,90,91]. IC4 isolates were also observed in European countries (Czech Republic, Germany, The Netherlands, Norway, Portugal, Spain, and Turkey [74,92]. The Norwegian isolate was, however, associated with an import from Pakistan [93]. This clone included outbreak, MDR, carbapenem-resistant (*bla_OXA-23_*_-like_-and *bla_OXA-58_*_-like_-positive), and *armA*-encoding isolates.

Isolates belonging to IC5 presented a broader spectrum of acquired antimicrobial resistance determinants compared to IC4. The presence of aminoglycoside-modifying enzymes encoding genes *aadA1*, *strA,* and *strB*, as well as *dfrA1,* has been previously associated with class 2 integrons in IC5 in different Latin American countries [94,95,96]. Moreover, *bla_TEM−1_* occurred in this IC more frequently and could contribute to the high ampicillin-sulbactam MICs observed, which were revealed in OXA-23-producing isolates in particular [55,97]. Mutations in the constitutive *pmrAB* genes could be responsible for the lower polymyxins susceptibility in IC5 isolates. Most amino acid substitutions were revealed in PmrB of major South American clones, especially among IC4 isolates, for which a rather high number of *pmrAB* mutations was observed. However, none of these mutations was unique to the isolates possessing polymyxin resistance. IC4 isolates were also characterized by the absence of the *eptA* gene [98].

IC5 is the prevalent clonal lineage found in Latin America, the so-called ‘Pan-American clone’, followed by IC4 [55,71,99]. However, both IC4 and IC5 were also reported in Europe and IC4–in Asia.

### 4.3. IC6

The distribution of IC6 isolates was described in European countries, Brazil, and the USA [74,99]. This genotype (ST78) was found for the first time in Italy in 2006 and then was identified in five patients from two additional hospitals in Naples in 2007 [100]. Italian IC6 isolates were resistant to all antimicrobials tested, including carbapenems, but were susceptible to colistin. They also possessed a plasmid-borne carbapenem-hydrolyzing oxacillinase gene, *bla_OXA-58_*, flanked by ISAba2 and ISAba3 elements.

The genomic characterization of 15 ST78 isolates from Italy in a recent investigation [101] revealed that they represented a monophyletic clade (named ST78A), which possessed a higher number of insertion sequences in comparison to other related clades (ST78B and ST49). A particular insertion sequence, IS66, was revealed to interrupt the gene *comEC/rec2* involved in the acquisition of exogenous DNA, thus limiting genome plasticity and explaining the low incidence of this clone according to the hypothesis formulated by the authors.

The strains of this genetic lineage are widespread in the territory of Russia and Belarus [77,102]. IC6 isolates from these neighboring countries were characterized by the highest frequency of resistance to different antimicrobial compounds, including colistin (designated as colistin- nonsusceptible *A. baumannii* isolates), as well as by the production of OXA-24/40 group carbapenemases [77,102]. Later, two IC6 isolates were investigated in Germany from patients injured in Russia and repatriated [103]. These isolates harbored the *bla_OXA-72_* (*bla_OXA-40_*_-like_) gene as a carbapenem resistance determinant and additionally possessed extended-spectrum beta-lactamase (ESBL) gene *bla_CTX-M-115_*. Later, the isolates of that genetic lineage were isolated from war-injured patients from Eastern Ukraine who were treated at German Bundeswehr Hospitals for humanitarian reasons during 2014–2015 [104]. These isolates were carbapenem- and fluoroquinolone-resistant and had an acquired *bla_OXA-72_* carbapenemase.

Thus far, IC6 was mostly reported in Russia, Belarus, Italy, and South American countries.

### 4.4. IC7

The first IC7 isolates were described in several South American countries, such as Paraguay and Argentina, but were usually sporadic [105]. IC7 isolates represented the most prevalent group in Bolivia and Uruguay [23], later suggesting a change in the epidemiology of carbapenem-resistant *A. baumannii* isolates in these countries [90]. Single XDR isolates of this genetic line were recently observed in Africa and Asia [106]. Two IC7 isolates were also identified among war-injured Ukrainian patients mentioned above [104], and an MDR isolate was recently reported in Russia [79]. The IC7 strains mainly harbored the *bla_OXA-23_* beta-lactamase gene. 

A comprehensive genomic analysis of nineteen IC7 isolates associated with outbreaks in different countries (Germany, Greece, The Netherlands, Italy, Slovenia, Sweden, Turkey, Singapore, Thailand, UAE, and Argentina) revealed the genotypic and phenotypic properties behind the worldwide distribution and evolution of this *A. baumannii* lineage [107]. According to the phylogenetic analysis of the IC7 (ST25) strains studied, multiple homoplasious genomic regions were identified, indicating active or historical recombination. Searching for differential gene conservation using the large-scale BLAST score ratio (LS-BSR) pipeline revealed a single coding region that corresponded to the hemagglutinin repeat protein, the complete gene structure of which is unique to ST25 genomes. The strains of this genetic lineage also demonstrated better adherence and a significantly higher biofilm formation in the absence of antimicrobials than the ST1 representatives.

Thus far, IC7 was reported in different regions of the world, including Europe, Asia, and North and South America.

### 4.5. IC8

IC8 *A. baumannii* clones were documented as single (sporadic) isolates in India [106], Afghanistan [108], Germany [109], Bangladesh [81], China [110], and the Philippines [111]. Multidrug resistance in *A. baumannii* from the investigated Bulgarian hospitals was also associated with intrahospital dissemination of IC8, along with IC1 and IC2 [14]. It is noteworthy to mention that there are some examples of IC8 strains isolated from dogs with common features among human and animal strains [112], and several avian isolates from livestock belonging to IC8 were also found [113]. Carbapenem resistance was mainly explained by the acquisition of the class-D β-lactamase gene *bla_OXA-23_* and intrinsic *bla_OXA-68_*. Interestingly, the carbapenemase gene *bla_NDM-6_* was found exclusively in the three IC8 genomes of the Philippine isolates. This rare metallo-beta-lactamase was first observed in Spain in an *A. baumannii* isolate belonging to IC9 [114].

Recently, a large fraction of IC8 isolates was reported in Europe, with sporadic reports from various Asian countries.

### 4.6. IC9

The most recent IC9 was previously identified in *A. baumannii* isolates recovered between 2012 and 2016 in Belgium (*bla_NDM-1_*-positive), Egypt (*bla_OXA-23_*-positive), Italy (*bla_NDM-1_*-positive), and Pakistan (*bla_OXA-23_*-positive) [68]. However, *A. baumannii* of ST85, which carried both *bla_OXA-94_* and *bla_NDM-1_*, were first reported in Lebanon in 2014, where they were isolated from the victims of the Syrian civil war [115]. In addition, the isolates encoding *bla_OXA-94_* and *bla_NDM-1_* were reported from Southern Spain, Saudi Arabia, and Tunisia, whereas the isolates harboring *bla_VIM-1_* were found in Egypt, which indicates that the novel IC9 clonal lineage has a widespread global distribution and usually includes metallo-beta-lactamases [64,116,117,118].

However, according to these reports, this genetic lineage is currently more specific to the Middle East and North Africa.

### 4.7. IC10 Candidate–CC33

The data presented above demonstrates that the clonal structure of *A. baumannii* cannot be fixed once and for all, and this bacterial species evolves under antibiotic pressure and the influence of other environmental factors. Some clones can gradually lose their importance, as observed for IC3, and new ICs can arise.

We have performed the analysis of all *A. baumannii* genomes available in GenBank and analyzed the ST structure using the eBURST algorithm. Based on the data obtained, we propose a new IC candidate, IC10, which was previously referred to as CC33. The partial eBURST output for Pasteur MLST is shown in Figure 3.

It can be seen that ST33 is the central element of this group. The distant and well-formed clusters can also be seen on the full tree of STs (see Appendix A). We analyzed the genomes of the isolates possessing STs clustered with ST33 for the presence of common OXA-51-like carbapenemases and found that the isolates belonging to ST33, ST132, ST151, ST193, ST213, ST235, ST424, ST448, ST456, ST459, ST911, and ST2034 possessed the same *bla_OXA-120_* gene, whereas the isolates of other STs from the cluster were not found in GenBank. The isolates of the corresponding STs were also clustered according to the cgMLST analysis.

In addition, all these isolates except one included the same variant of the intrinsic gene *bla_ADC-25_*, namely, *bla_ADC-156_*. The antibiotic resistance gene content for the isolates was low; only one isolate contained *bla_OXA-23_* and the other contained *bla_NDM-1_*. Nevertheless, *A. baumannii* has been shown to rapidly acquire resistance determinants several times, and thus CC33 should not be underrated.

According to the data from PubMLST [46] (accessed on 29 March 2023), the clinical isolates of CC33 were revealed in China, Japan, Saudi Arabia, Poland, Russia, Brazil, and the USA during 2005–2021. Previously, the members of CC33 were also reported in Bangladesh (ST459) [81], Brazil (ST151) [88], and Lebanon (ST284) [119]. Thus, CC33 has already spread across continents including Europe, Asia, and North and South America. Some reports of non-clinical CC33 obtained from animals were also published [14,67].

One of the CC33 isolates (CriePir309, which was uploaded to GenBank under GCA_016654295 and belonged to ST911) was previously revealed in a Russian hospital and sequenced by us [79]. Notably, it was susceptible to all antibiotics in the panel.

We can conclude that CC33 fulfills the criteria for its assignment to novel international clones, but currently, antibiotic resistance for the isolates of this IC10 candidate is rather low.

## 5. Distribution and Characteristics of ICs Available in Genbank

Isolate typing is crucial for conducting epidemiological surveillance and investigating the spread of particular clones of interest, including ICs. The rapid development and dramatic cost reduction of WGS for bacteria in recent years have made genomic epidemiology a promising approach for pathogen population studies and outbreak investigation [20,120,121,122,123].

Currently, more than 17,000 complete and partial genomes are available in GenBank databases for *A. baumannii*, but in most cases, they do not have the accompanying epidemiological data, including STs or other typing results. The PubMLST database (https://pubmlst.org/organisms/acinetobacter-baumannii, accessed on 25 May 2023) includes epidemiological data (ST, OXA), but the number of genomes is about 3200, and cgMLST data are usually not available. Another advantage of this database is that the species-level identification provided in it was validated by both MLST and ribosomal MLST typing. This ensures the correct species assignment and eliminates possible errors when the isolates belonging to other Acinetobacter species (e.g., *A. pittii*) could be incorrectly assigned to *A. baumannii* based on partial genomic sequences only. However, a comprehensive investigation of some clinical sets of the isolates requires comparison with available data for particular STs, and searching for the isolates with similar profiles requires substantial computational experience. In addition, the attribution of the reference isolates to particular ICs is not always straightforward, as discussed above.

A number of bioinformatics tools have been developed for performing genomic sequence analysis of bacterial pathogens, including *A. baumannii*. These tools include, among others, BacWGSTdb [124], which integrates data from various sources, including NCBI, PubMLST, ResFinder, and VFDB, and also provides computational tools for analyzing whole genomes uploaded by users; Pathogenwatch (https://pathogen.watch/, accessed on 30 March 2023), which also integrates annotation data and provides visualization tools for various pathogens, including *A. baumannii*; 5NosoAE [125], that is targeted to nosocomial bacterial antibiogram investigation. However, none of these tools provides the possibility of straightforward and unambiguous assignment of a particular isolate to known ICs.

In order to facilitate future genomic epidemiology investigations and to obtain a snapshot of sequenced *A. baumannii* isolates, we have analyzed the whole set of genomes available in GenBank (accessed on 14 January 2023). A total of 17,546 sequences were retrieved, for which the assembly level was either ‘Complete Genome’, ‘Chromosome’, or ‘Scaffold’. It was already observed that the set of genomes available in GenBank is not representative and is strongly biased towards clinically important (e.g., outbreak or MDR) clones with ST2, ST1, ST79, and ST25 accounting for over 71% of the isolates [54]. However, the goal of the analysis was not to perform a comprehensive population study, but to provide useful reference data for future investigations and to illustrate the IC distribution on real data.

We determined the STs and the presence of OXA-51-like β-lactamase encoding genes, calculated cgMLST profiles, and assigned ICs, where possible. We used an extended version of the bacterial genome analysis pipeline described earlier [126,127] by us.

Resfinder 4.3.0 software was used for antimicrobial gene detection (https://cge.cbs.dtu.dk/services/ResFinder/, accessed on 20 February 2023, using default parameters), including OXA-51-like variant typing. MLST typing was performed using the PubMLST database (https://pubmlst.org/bigsdb?db=pubmlst_abaumannii_seqdef, accessed 14 February 2023). The cgMLST profiles were built using MentaList (https://github.com/WGS-TB/MentaLiST, version 0.2.4, default parameters, accessed on 25 May 2023) [52] using the scheme obtained from cgmlst.org (https://www.cgmlst.org/ncs/schema/schema/3956907/, accessed on 30 March 2023, contained 2390 loci, last update 20 February 2023 [29]).

The complete data for MLST and OXA-51-like typing are shown in Appendix A, and the cgMLST profiles are shown in Appendix A for all isolates. The data are also available on Github (https://github.com/fallandar/bacteria_typing, accessed on 30 March 2023), where the updates will be provided in the future. The data shown in the table can be used in future investigations as a reference for comparison purposes. For example, researchers can easily compare their isolates with those from GenBank using cgMLST profiles without the necessity of performing long computations.

In the section below, we have referred to ‘isolate’ while discussing a GenBank assembly record containing a complete or partial genome for simplicity, although some distinct records could represent the same isolate or some part of the isolate genome.

In total, 482 distinct STs were found in 17,546 complete or partial genomes of *A. baumannii* isolates, of which ST2 was detected in 63.3% (11,108) of the cases. In contrast, the second-largest ST1 was found only for 3.6% of the isolates. Overall, 78.5% of the isolates belonged to ICs. A summary of the number of isolates assigned to ICs based on their ST and OXA-51-like type (see Table 4) is shown in Figure 4.

We also analyzed the presence of some beta-lactamases associated with increased resistance in *A. baumannii*–*bla_OXA-23_*, *bla_OXA-24-_*_like_ (including *bla_OXA-24/40_* and *bla_OXA-72_*), bla_OXA*-58*_, *bla_NDM-_*_1_, *bla_NDM-6_*, and *bla_CARB_*_._ *Bla_OXA-23_* was detected in 95% of the IC2 isolates, which agrees with a previous study [128], and in 90% of the IC4 isolates, but it was found in a relatively low fraction of the isolates not belonging to IC (40%), as well as in IC6 (1%) and IC9 (6%). However, as it was mentioned above, the GenBank *A. baumannii* population is biased towards multidrug-resistant clinical isolates, and thus any conclusions except for the purely observational ones seem to be unreliable.

*Bla_OXA-_*_58_ was found in all ICs, with a slight predominance in IC5, whereas *bla_NDM-_*_1_ was mainly found in IC2 and IC1 and absent in IC3 and IC4. Notably, *bla_NDM-6_* was mainly found in IC8 and IC9 isolates (8 of 10 cases), as revealed previously (see Section 4.5 and Section 4.6), but was also detected in IC1 (2 isolates). In contrast, *bla_CARB_* genes were predominantly found in the isolates not belonging to any ICs (218 cases), followed by IC6 and IC1 (34 and 28, respectively). *Bla_OXA-_*_24-like_ genes were revealed in all ICs except IC9, with a relative predominance in IC5, where they were found in about half of all isolates.

A summary of all antimicrobial resistance genes identified in *A. baumannii* isolates from GenBank is presented in Appendix A.

In general, IC1 and IC2 showed higher resistance than other ICs and isolates that do not belong to any IC, but more careful investigations are needed to deduce reliable comparison results.

## 6. Misconceptions, Challenges, and Clarifications of IC Definitions

The main goal of this review, as we stated above, was to provide researchers with up-to-date definitions of ICs and to elucidate the possible difficulties in performing IC assignments for recently sequenced *A. baumannii* isolates. The process of such an assignment is not always straightforward, which can produce inconsistencies in the data obtained by different research groups.

In particular, the eBURST algorithm [58], which is widely used for clustering STs, can output different results for particular organisms from time to time when more STs become available in public databases. As we mentioned above, the number of Pasteur scheme-based STs increased from 59 to more than 2000 from 2010 to 2023, and researchers continue to upload novel STs when they reveal previously unknown MLST allele combinations in their isolates. Thus, the composition of the clusters used for computational IC definition, as well as the central elements of such clusters, could also change.

Another issue is that the set of reported STs is biased towards widespread clinical clones, and some intermediate data useful for cluster building could be missing since the isolates of less frequent STs have not yet been uploaded to the databases.

For example, the central element of the IC3 cluster was defined as ST3 [13], and thus the corresponding cluster was referenced as CC3. This concept is currently used in most cases [86,129,130], but some other researchers also consider ST124 for this role [14,84], thus defining CC124.

Hamidian et al. [47] pointed out one more important problem for CC92^Oxf^ from the Oxford MLST scheme, which was supposed to represent IC2. The authors revealed that ST92^Oxf^ was highly likely to represent an artifact caused by an erroneous primer sequence used for the *gpi* gene, and the correct type, in this case, should have been ST208^Oxf^. In addition, ST109^Oxf^, a commonly reported ST for IC1 isolates, was in fact ST231^Oxf^. These erroneous STs can still be found in some databases since they were uploaded previously and updates were not provided. Additionally, some researchers still use CC92^Oxf^ and CC109^Oxf^ in their investigations (e.g., [84,131]), which introduces confusion for readers.

The classification based on *bla_OXA-51_*_-like_ gene variants is also prone to some challenges. First, although this gene family was considered intrinsic to *A. baumannii*, several reports revealed *bla_OXA-51_*_-like_ genes in another Acinetobacter, namely, Acinetobacter genomic species 13TU [132], and in the members of *Enterobacteriaceae*, namely, *Enterobacter cloacae* and *Klebsiella pneumoniae* [133]. Second, novel OXA-51 variants can be uploaded to dedicated databases (e.g., [69]) in the same way as novel STs, and some of them might be found in the isolates belonging to ICs. Thus, the list of variants attributed to particular ICs should be updated on a regular basis. The influence of these problems can be significantly lowered when *bla_OXA-51_*_-like_ typing is used in combination with some other technique(s) like MLST.

Possible problems with cgMLST include significant computational skills required to correctly deduce the IC information based on corresponding profiles, as well as possible low sequencing quality, which could prevent the successful calling of cgMLST alleles. In contrast, MLST allele calling is less sensitive to sequencing quality [134].

In general, the approaches based on the combination of MLST, *bla_OXA-51_*_-like_, and when available and suitable, cgMLST typing seem to be much more reliable than those involving single techniques or purely molecular approaches like DiversiLab.

In Table 3 and Table 4 of this review, we provide the verified properties of ICs currently used, including both MLST schemes and OXA-51 variants, which, in our opinion, will facilitate the standardization of such definitions.

## 7. Conclusions

In this review, we discussed the history, definitions, spread, and genomic data available for *A. baumannii* international clones of high risk 1–9. We also discussed the approaches and provided data for IC identification based on genomic sequences (in particular, Table 4), which are thought to be known a priori, but can hardly be found in one place when needed. The possible emergence of a novel clone, IC10, also known as CC33, was considered.

We believe that this review will facilitate the epidemiological and genomic investigations of *A. baumannii*, especially for researchers who have recently started their studies in the promising field of genomic epidemiology.

## Figures and Tables

**Figure 1 microorganisms-11-02115-f001:**
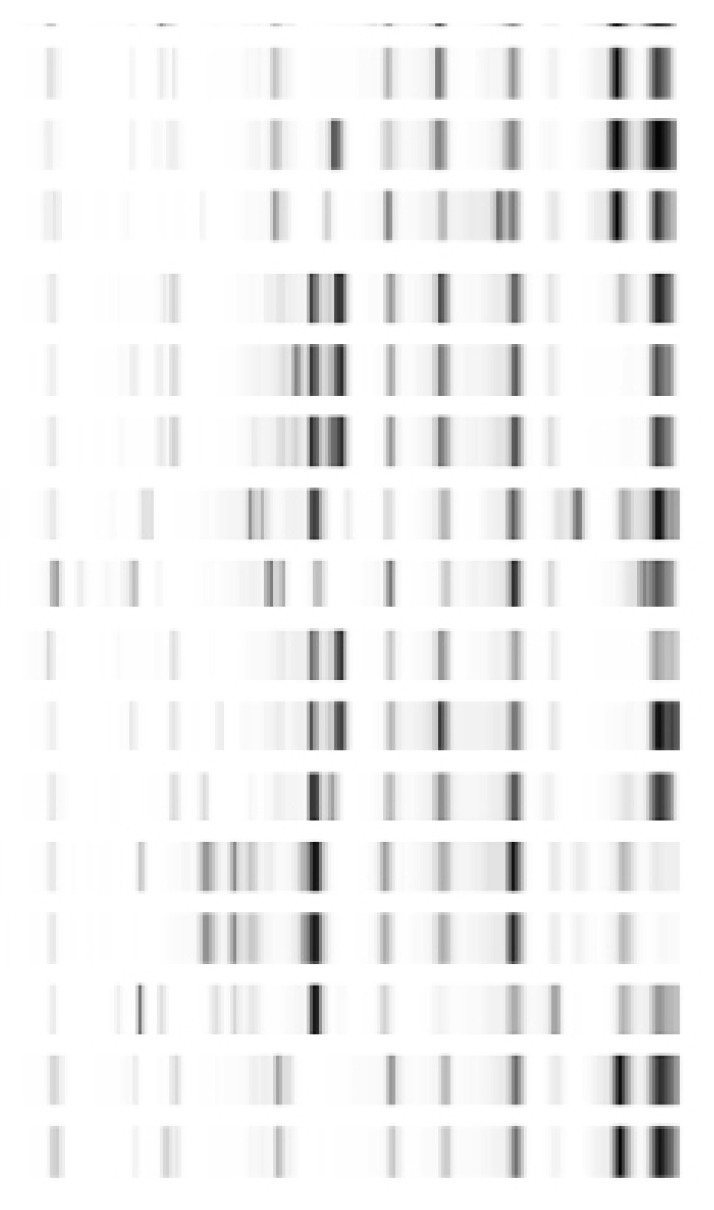
An example of rep-PCR profiles. Although some similar profiles can be seen, computer-aided classification seems to be a better option.

**Figure 2 microorganisms-11-02115-f002:**
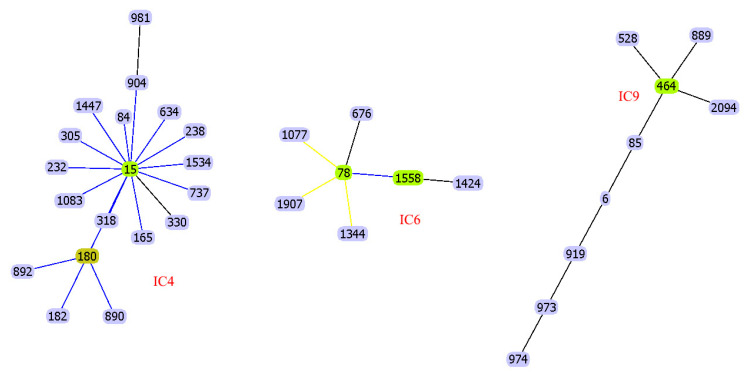
Classification of *A. baumannii* Pasteur STs for IC4, IC6, and IC9 using the eBURST algorithm. Blue links show single-locus variants (SLVs). Founder STs are highlighted in yellow.

**Figure 3 microorganisms-11-02115-f003:**
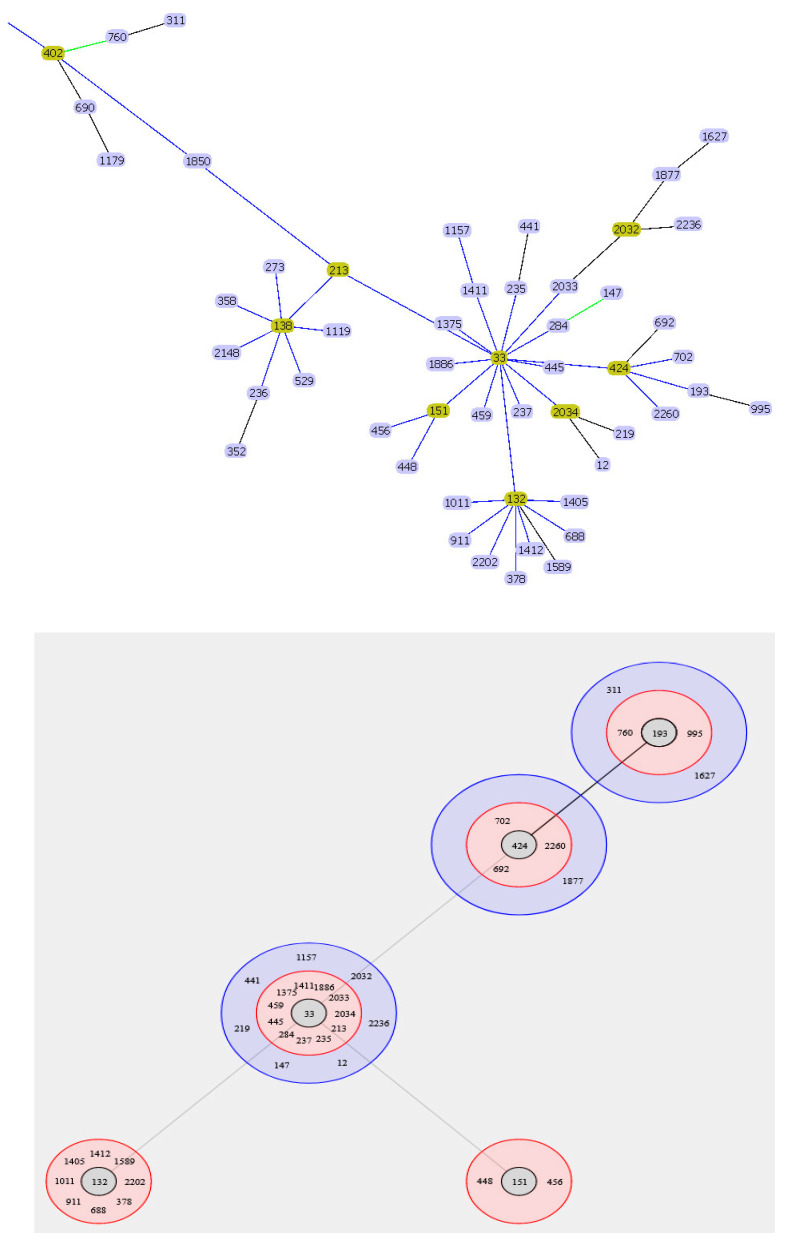
eBURST analysis of CC33–IC10 candidate. (**top** panel) eBURST tree structure. Blue links show single-locus variants (SLVs). Founder STs are highlighted in yellow. (**low** panel) Circular eBURST diagram. The pink circle includes SLVs for ST in the center, while the blue circle contains STs that differed by 2 MLST loci from the central ST.

**Figure 4 microorganisms-11-02115-f004:**
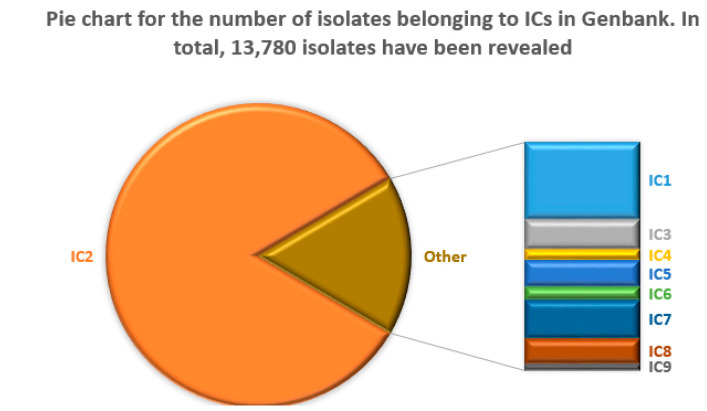
Pie chart showing the number of isolates belonging to ICs in GenBank. Only the data for the isolates assigned to ICs are shown.

**Table 1 microorganisms-11-02115-t001:** The methods of isolate and clone typing for *A. baumannii*, which could possibly be used for IC assignment.

Typing Method	Group	Target	Results	Advantages	Limitations	Ref.
Amplified fragment length polymorphism (AFLP)	DNA banding pattern	Whole genome	50-band pattern	Deep resolution, high discriminatory power (DP)	Laborious, expensive, low inter-reproducibility	[22]
Repetitive sequence based-PCR (rep-PCR)	DNA banding pattern	Whole genome	Band pattern	Rapid, high DP	Expensive	[23]
Pulsed-field gel electrophoresis (PFGE)	DNA banding pattern	Whole genome	20-band pattern	High DP	Laborious, low inter-reproducibility	[24]
Trilocus sequence-based typing (3-LST)	DNA banding pattern, Allele sequencing	Polymorphism within *ompA*, *csuE* and *bla_OXA-51_*		Rapid, easy, public database availability	Inability to recognize certain ICs and non-IC isolates	[25]
Multilocus sequence typing (MLST)	Allele sequencing	Polymorphism within seven housekeeping genes	Sequence type	Portable, reproducible, public database availability	Expensive, limited availability in clinical settings	[13,26]
*bla_OXA-51_*-_like_ gene	Allele sequencing	Polymorphism within one locus	OXA variant	Cheap, public database availability	Low specificity	[27]
Capsule synthesis loci (KL) and lipooligosaccharide outer core loci (OCL)	Allele or whole genome sequencing	Polymorphism within KL and OCL	KL and OCL types	Reproducible, public database availability	Low DP, expensive, limited availability in clinical settings	[28]
cgMLST	Whole genome sequencing	Whole genome polymorphism	2390 gene allelic profile	Rapid, automated, provides a lot of various data, high DP	Expensive, limited availability in clinical settings	[29]
core genome Single Nucleotide Polymorphisms (SNP)	Whole genome sequencing	Whole genome polymorphism	SNP-based profile	Rapid, provides a lot of various data, high DP	Expensive, limited availability in clinical settings	[30,31]

**Table 2 microorganisms-11-02115-t002:** Exemplary Pasteur MLST-based sequence type (ST) for *A. baumannii* isolate. Numbers represent the variants of the corresponding alleles. ST is uniquely determined by the combination of these seven numbers, and the definitions are stored in public databases.

Isolate ID	*cpn60*	*fusA*	*gltA*	*pyrG*	*recA*	*rplB*	*rpoB*	ST
Exp1	3	29	30	1	9	1	4	17

**Table 3 microorganisms-11-02115-t003:** The common attributes of international clones (IC) *of A. baumannii*.

	Pas *	Oxf	Major OXA-51 Variant	Minor OXA-51 Variants **
IC1	CC1	CC231	OXA-69	OXA-92, OXA-107, OXA-110, OXA-112
IC2	CC2	CC208, CC218, CC281	OXA-66	OXA-82, OXA-83, OXA-84, OXA-109, OXA-172, OXA-201, OXA-202
IC3	CC3	CC106	OXA-71	OXA-113
IC4	CC15	CC103	OXA-51	OXA-98
IC5	CC79	CC205	OXA-65	-
IC6	CC78	CC944	OXA-90	OXA-200
IC7	CC25	CC229	OXA-64	-
IC8	CC10	CC447	OXA-68	OXA-128, OXA-144
IC9	CC464	CC1078	OXA-94	-

* Pas shows Pasteur MLST scheme [13] and Oxf–Oxford scheme [26]. ** Minor variants were deduced using experimental data from the literature [62,65] and bioinformatics analysis of *A. baumannii* genomes from GenBank was performed by us (details are provided in Section 5).

**Table 4 microorganisms-11-02115-t004:** The list of MLST-based STs and *bla_OXA-51_*_-like_ gene variants associated with particular IC.

IC	Pasteur Scheme Sequence Types (ST)	OXA-51 Variants
IC1	1, 19, 20, 81, 94, 315, 460, 623, 717, 734, 986, 1090, 1106	OXA-69, OXA-92, OXA-107, OXA-110, OXA-112
IC2	2, 45, 187, 195, 414, 492, 570, 571, 577, 600, 604, 664, 745, 823, 1537, 1550, 1555, 1579	OXA-66, OXA-82, OXA-83, OXA-84, OXA-109, OXA-172, OXA-201, OXA-202
IC3	3, 124, 229, 500, 1822	OXA-71, OXA-113
IC4	15, 238	OXA-51, OXA-98
IC5	79, 156, 175, 422, 730, 1163, 1196	OXA-65
IC6	78	OXA-90, OXA-200
IC7	25, 113, 619, 945, 1487	OXA-64
IC8	10, 23, 82, 575, 613, 1512	OXA-68, OXA-128, OXA-144
IC9	6, 85, 464	OXA-94

## Data Availability

The MLST and cgMLST data discussed in the manuscript are available as Appendix A and in Github repository https://github.com/fallandar/bacteria_typing (accessed on 31 May 2023, will be updated in the future).

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
