# Peer review of "International Clones of High Risk of Acinetobacter Baumannii—Definitions, History, Properties and Perspectives"

_microorganisms, 2023, doi:10.3390/microorganisms11082115_

Round 1

Reviewer 1 Report

Manuscript number 2349877 submitted to Microorganisms is a potentially interesting review on the International High Risk clones of Acinetobacter baumannii. The objective of the review is to analyze “A. baumannii typing tools suitable for IC determination, provide the data to easily determine IC assignment based on MLST sequence type (ST) and intrinsic blaOXA-51-like gene variants..MLST and cgMLST profiles, OXA-51-like presence data will be given for all isolates available in Genbank”. Unfortunately, the study have several major errors and technical mistakes, which hinder to provide a clear and correct analysis of genomic epidemiology of A. baumannii. I am listing below specific queries which have to be addressed by Authors.

1.       The major concern I have with the study is that Authors use the semi-automated (rep)-PCR DiversiLab (bioMérieux, Marcy l’Etoile, France) technique to identify and classify international clonal lineages. Although the DiversiLab typing method has been often employed to screen for the presence of distinct epidemic clones among A. baumannii isolates, this technique needs to be confirmed by MLST typing. This is clearly stated in the original publication on the use of DiversiLab technique to genotype A. baumannii isolates: Higgins et al. J Clin Microbiol. 2012 Nov;50(11):3493-500. doi: 10.1128/JCM.01759-12. Based on the above major concern, it my suggestion that Authors identify and classify A. baumannii international clones according to Pasteur MLST scheme’s nomenclature as described in references 13 and 44.

2.       Table 1. MALDI-TOF is an identification method buti t is not a typing method. The correct reference for the use of MOLDI-TOF in the identification of species into A. baumannii group is the following: Marí-Almirall et al. Clin Microbiol Infect. 2017 Mar;23(3):210.e1-210.e9. doi: 10.1016/j.cmi.2016.11.020. 

3.       Table 1. Single-locus based typing using blaOXA-51 like gene. According to the original publication describing the method by Pournaras et al. J Clin Microbiol 2014, 52, 1653-1657, which is quoted in the manuscript as reference 63, the complete DNA sequence of blaOXA-51 like gene is needed to identify alleles and variants. The reference 26 is misquoted for this method.

4.       Table 1. KL locus and OCL locus typing possesses an high discriminatory power, not a low discriminatory power.

5.       Table 1. In addition to the Pasteur MLST scheme, Authors need to consider the Oxford MLST scheme also.

6.       Table 1. The tri-locus-based typing method descibed by Turton et al. Clin Microbiol Infect 2007; 13:807–15 should be included also.

7.       Table 1. Authors need to consider whole genome sequencing based typing methods such as SNP-based methods other than core genome MLST method as recently described by Petazzoni et al. Microbiol Spectr. 2023 Mar 23:e0450522. doi: 10.1128/spectrum.04505-22.

8.       Table 3. The nomenclature of clonal lineages according to Oxford MLST scheme is wrong and does not consider informations reported in references 44 and 45 of the study. For example, Oxford CC92 is a PCR typing artifacts and does not exists among A. baumannii genomes. Morerover, the higher discriminatory power of Oxford MLST scheme compared with Pasteur MLST scheme need to be shown.

9.       The publication by Gaiarsa et al. Front Microbiol. 2019 Sep 12;10:2080. doi:10.3389/fmicb.2019.02080 should be quoted and discussed for the genomic characterization of Pasteur ST78 clonal lineage.

10.   Results. Section 5, Distribution and characteristics of ICs available in Genbank. In ther first paragraph, Authors should consider that the identification at species level for the genome collection of Acinetobacter baumannii genomes reported at PubMLST database (https://pubmlst.org/organisms/acinetobacter-baumannii) has the advantage to be validated by MLST typing  and ribosomal MLST typing. This is important for A. baumannii because there are several non-baumannoii Acinetobacter genomes which have been erroneously depositeds as A. baumannii in GenBank.

11.   Core genome MLST reluts, main text lines 527-530 and Table S2. The number of high quality genes used to define core genome of A. baumannii should be indicated and results included in Table S2 explained.Alternatively, I would suggest to eliminate this part.

12.   ResFinder asnalyses of A. baumannii genomes. I was surprised that blaOXA24/40 or blaOXA72 CHDLs genes were not found among A. baumannii genomes.

No comments.

Author Response

Manuscript number 2349877 submitted to Microorganisms is a potentially interesting review on the International High Risk clones of Acinetobacter baumannii. The objective of the review is to analyze “A. baumannii typing tools suitable for IC determination, provide the data to easily determine IC assignment based on MLST sequence type (ST) and intrinsic blaOXA-51-like gene variants..MLST and cgMLST profiles, OXA-51-like presence data will be given for all isolates available in Genbank”. Unfortunately, the study have several major errors and technical mistakes, which hinder to provide a clear and correct analysis of genomic epidemiology of A. baumannii. I am listing below specific queries which have to be addressed by Authors.

We would like to thank the reviewer for a thorough analysis of our manuscript and for very useful suggestions, which has led to significant improvements in the manuscript readability. Our answers to the comments are provided below bold-faced.

Before answering the comments, we would like to make clear the goal of this review. It is dedicated not to genomic epidemiology of A. baumannii in general, but rather to providing definitions and the methods of assigning the isolates to several clonal lineages known as international clones of high risk (IC). Such definitions are usually made using cluster analysis, and sometimes are used in a controversial way. We would like to provide the readers, which could possibly be new to the field, with a comprehensive description of possible IC definitions and describe the procedure of assigning a particular isolate to known ICs.  Thus, the description of typing methods includes only the ones that can possibly be applied to this purpose, while other approaches not considered in the review could be suitable, for example, for outbreak investigation or other purposes. Genbank analysis was performed just for illustration purposes since this genomic set cannot be considered as representative for the whole population of A. baumannii in the world.

  1. The major concern I have with the study is that Authors use the semi-automated (rep)-PCR DiversiLab™ (bioMérieux, Marcy l’Etoile, France) technique to identify and classify international clonal lineages. Although the DiversiLab typing method has been often employed to screen for the presence of distinct epidemic clones among A. baumannii isolates, this technique needs to be confirmed by MLST typing. This is clearly stated in the original publication on the use of DiversiLab technique to genotype A. baumannii isolates: Higgins et al. J Clin Microbiol. 2012 Nov;50(11):3493-500. doi: 10.1128/JCM.01759-12. Based on the above major concern, it my suggestion that Authors identify and classify A. baumannii international clones according to Pasteur MLST scheme’s nomenclature as described in references 13 and 44.

Our manuscript is a review paper, so we have not performed any molecular experiments, and thus we have not used DiversiLab™ technique in any way. We just described the techniques, methods and approaches used previously and currently to identify international clones of high risk. DiversiLab™ historically was the first method to define ICs, so we described its application. We clearly stated in the text that MLST and cgMLST (when available) are the most modern and precise methods of identification, although  some researchers still use DiversiLab™ or similar techniques in some circumstances for this purpose (mainly, due to its low cost in comparison to WGS) (see the concluding remarks for section 2 and description in section 3) .

We used cgMLST, MLST and OXA-51-like variant analysis for assigning A. baumannii isolates available in Genbank to particular ICs (section 5).

  1. Table 1. MALDI-TOF is an identification method buti t is not a typing method. The correct reference for the use of MOLDI-TOF in the identification of species into A. baumannii group is the following: Marí-Almirall et al. Clin Microbiol Infect. 2017 Mar;23(3):210.e1-210.e9. doi: 10.1016/j.cmi.2016.11.020. 

Although MALDI-TOF is usually used as the identification method, some researchers described its possible applicability to strain typing (e.g., Spinali S, van Belkum A, Goering RV et al. Microbial typing by matrix-assisted laser desorption ionization-time of flight mass spectrometry: do we need guidance for data interpretation? J. Clin. Microbiol. 53(3), 760–765 (2015); Mencacci A, Monari C, Leli C et al. Typing of nosocomial outbreaks of Acinetobacter baumannii by use of matrix-assisted laser desorption ionization-time of flight mass spectrometry. J. Clin. Microbiol. 51(2), 603–606 (2013);  Rim JH, Lee Y, Hong SK et al. Insufficient discriminatory power of matrix-assisted laser desorption ionization time-of-flight mass spectrometry dendrograms to determine the clonality of multi-drug-resistant Acinetobacter baumannii isolates from an intensive care unit. Biomed. Res. Int. 2015, 535027 (2015); Ghebremedhin M, Heitkamp R, Yesupriya S, Clay B, Crane NJ. Accurate and rapid differentiation of Acinetobacter baumannii strains by raman spectroscopy: a comparative study. J. Clin. Microbiol. 55(8), 2480–2490 (2017)).

Our goal was to describe all methods that could possibly be used for A. baumannii assignment to ICs, in particular, which represents the typing in a broader sense. We clearly stated and discussed the disadvantages of this method for the stated purpose. Thus, we gave reference to the paper describing such an application, not to the paper describing A. baumannii identification.

We have changed the title of Table 1 to elucidate the purpose of describing the methods and approaches contained in it.

  1. Table 1. Single-locus based typing using blaOXA-51 like gene. According to the original publication describing the method by Pournaras et al. J Clin Microbiol 2014, 52, 1653-1657, which is quoted in the manuscript as reference 63, the complete DNA sequence of blaOXA-51 like gene is needed to identify alleles and variants. The reference 26 is misquoted for this method.

Thank you for pointing this issue out. The reference has been replaced as suggested.

  1. Table 1. KL locus and OCL locus typing possesses an high discriminatory power, not a low discriminatory power.

KL and OCL loci possess low discriminatory power in comparison to MLST since the number of available KL types and OCL types is 237 and 22, respectively, which is much lower than the number of available STs (more than 1000). Again, we should state that we described not the methods for isolate typing in general (which was described in some reviews before, e.g., 10.2217/fmb-2019-0134), but to their assignment to nine known international clones, which clearly represents a different task. Since currently no clear correlation between KL-OCL types and ICs was revealed, these typing techniques cannot be used alone for the purpose of IC assignment.

  1. Table 1. In addition to the Pasteur MLST scheme, Authors need to consider the Oxford MLST scheme also.

In Table 1 we reported MLST typing in general without specifying particular scheme. The references are now provided to both schemes

  1. Table 1. The tri-locus-based typing method descibed by Turton et al. Clin Microbiol Infect 2007; 13:807–15 should be included also.

Thank you for the suggestion. This method has been added to the table

  1. Table 1. Authors need to consider whole genome sequencing based typing methods such as SNP-based methods other than core genome MLST method as recently described by Petazzoni et al. Microbiol Spectr. 2023 Mar 23:e0450522. doi: 10.1128/spectrum.04505-22.

The SNP-based methods described in the reference given are used to perform fine typing, e.g., to reveal outbreak isolates. They are not very useful for IC assignment since they go into too much detail. Although they possess increased resolution power, they were not used to IC assignment in the investigations of other authors, which we studied, and thus are not suitable to be described in the current review.

  1. Table 3. The nomenclature of clonal lineages according to Oxford MLST scheme is wrong and does not consider informations reported in references 44 and 45 of the study. For example, Oxford CC92 is a PCR typing artifacts and does not exists among A. baumannii genomes. Morerover, the higher discriminatory power of Oxford MLST scheme compared with Pasteur MLST scheme need to be shown.

Although Oxford MLST possessed higher discriminatory power than Pasteur MLST, it suffers from severe problems imposed by the presence of a second copy of the Oxford gdhB locus genomes that has led to the creation of artefactual profiles and STs (10.3389/fmicb.2019.00930). According to this study, the Pasteur scheme appears to be less discriminant among closely related isolates, but less affected by homologous recombination and more appropriate for precise strain classification in clonal groups, which within this scheme are more often correctly monophyletic. The Oxford scheme also has the following important issues: gdhB paralogy, recombination, primers sequences, position of the genes on the genome.

As it was stated in another work (https://doi.org/10.1128/JCM.00533-17), the Pasteur scheme more readily identifies members of clonal complexes (and ICs, in particular), the Oxford scheme better reveals the diversity in members of the same clone in the region of the capsule locus. Since the goal of this study was to perform IC assignment, we mainly use Pasteur scheme and reported possible Oxford STs for information purposes only. For example, when we analyzed Genbank A. baumannii genomes, we were unable to determine Oxford-based ST for most genomes due to presence of two gdh copies.

We corrected the nomenclature for Oxf scheme in the Table 3, but we have not relied on it in IC assignment and provided it for illustration only since it is still used in some investigations for this purpose. Since the existing references are somewhat contradictory, we have performed additional eBurst analysis to verify Oxford CC assignments.

  1. The publication by Gaiarsa et al. Front Microbiol. 2019 Sep 12;10:2080. doi:10.3389/fmicb.2019.02080 should be quoted and discussed for the genomic characterization of Pasteur ST78 clonal lineage.

We added the discussion of this paper to section 4.3.

  1. Results. Section 5, Distribution and characteristics of ICs available in Genbank. In ther first paragraph, Authors should consider that the identification at species level for the genome collection of Acinetobacter baumannii genomes reported at PubMLST database (https://pubmlst.org/organisms/acinetobacter-baumannii) has the advantage to be validated by MLST typing and ribosomal MLST typing. This is important for A. baumannii because there are several non-baumannoii Acinetobacter genomes which have been erroneously depositeds as A. baumannii in GenBank.

Thank you for the useful suggestion. We have added the statement regarding PubMLST in this paragraph.

  1. Core genome MLST reluts, main text lines 527-530 and Table S2. The number of high quality genes used to define core genome of A. baumannii should be indicated and results included in Table S2 explained.Alternatively, I would suggest to eliminate this part.

We have added the information to the Table S2. 2390 high quality loci were used to define this scheme, and this information is given in the paragraph. The data provided in Table S2 can be used by researchers for comparison purposes, namely, to quickly find the appropriate reference genomes for the isolates they study. Thus, this table does not have a scientific value itself, but rather provide a very useful reference tool to reduce computations in future investigations, and these advantages supplement the similar isolate information provided by other genome-based databases. We have mentioned this in the manuscript.

  1. ResFinder asnalyses of A. baumannii genomes. I was surprised that blaOXA24/40 or blaOXA72 CHDLs genes were not found among A. baumannii genomes.

Thank you for pointing this out. These genes were in fact revealed in the genomes, but we have not reported them, although these genes are important for resistance acquisition. We added the corresponding data to the section 5.

Reviewer 2 Report

The manuscript is written clearly and comprehensively. The topic is adequately covered and supported by appropriate references and illustrations. The quality of work would be improved if a recent update was added in the post-COVID-19 period.

The authors should check language and grammar using reliable software.

Author Response

We thank the reviewer for suggestion. The results for the genomes uploaded to public databases in the post-COVID-19 period have been already included to the manuscript. Although resistance and virulence characteristics of A. baumannii could change during COVID-19 pandemic, the level of such differences is still debated, and, to the best of our knowledge, the differences revealed did not affect the known international clone grouping.

Reviewer 3 Report

This review paper firstly summarized the conventional and most advanced molecular typing methods for A. baumannii. Then they analyzed more than 17,000 A. baumannii genomes retrived from NCBI GenBank database to detect their MLST-based sequence types, carriage of OXA-51 variants, and cgMLST profiles. Finally, the prevalence and global spread of nine high-risk international clones were briefly discussed. Overall, this review paper is clearly presented and well written, which provides useful information for those who have interest in the study of genomic epidemiological characteristic of A. baumannii. Here, I would like to offer some suggestions.
1. Could you also add a paragraph for the summarize of the currently available bioinformatics tools for the genomic sequence analysis of A. baumannii. For example, BacWGSTdb, Pathogenwatch, and 5NosoAE.
2. As you have performed a cgMLST analysis for all A. baumannii isoaltes, please also include a minimum spanning tree and highlight all the nine international clones mentioned in the review.
3. Please also include the results of antimicrobial resistance genes (especially carbapenem resistance genes) and KL and OCL information for all A. baumannii isoaltes.
4. Please provide a phylogenetic tree for all mentioned OXA-51 variants in Table 3.
5. Please remove Figure 1.
6. Please correct the typos in Table 1. For example, BlaOXA-51-like gene should be blaOXA-51-like gene, CgMLST should be cgMLST.

Please carefully proofread the manuscript to check all the typos and grammatical errors.

Author Response

This review paper firstly summarized the conventional and most advanced molecular typing methods for A. baumannii. Then they analyzed more than 17,000 A. baumannii genomes retrived from NCBI GenBank database to detect their MLST-based sequence types, carriage of OXA-51 variants, and cgMLST profiles. Finally, the prevalence and global spread of nine high-risk international clones were briefly discussed. Overall, this review paper is clearly presented and well written, which provides useful information for those who have interest in the study of genomic epidemiological characteristic of A. baumannii. Here, I would like to offer some suggestions.

We would like to thank the reviewer for a thorough analysis of our manuscript and for very useful suggestions, which has led to significant improvements in the manuscript readability. Our answers to the comments are provided below bold-faced.

  1. Could you also add a paragraph for the summarize of the currently available bioinformatics tools for the genomic sequence analysis of A. baumannii. For example, BacWGSTdb, Pathogenwatch, and 5NosoAE.

The paragraph was added to section 5.

  1. As you have performed a cgMLST analysis for all A. baumannii isoaltes, please also include a minimum spanning tree and highlight all the nine international clones mentioned in the review.

A minimum spanning tree for 17,000 isolates is bulky and not suitable for human reading. Due to significant bias towards IC2 and IC1 in Genbank such a tree will not be representative either, and can appear misleading for the readers. We showed the ICs in eBURST trees (Figure 2, Figure S1 – general picture), which are much more informative in our opinion

  1. Please also include the results of antimicrobial resistance genes (especially carbapenem resistance genes) and KL and OCL information for all A. baumannii isoaltes.
    These data was added to Tables S1 and S2.
  2. Please provide a phylogenetic tree for all mentioned OXA-51 variants in Table 3.

The tree is now provided in Figure S2.

  1. Please remove Figure 1.

We believe that this Figure will be useful for the readers not familiar with PFGE technique and other similar methods to get the impression of what the results of these approaches could look like

  1. Please correct the typos in Table 1. For example, BlaOXA-51-like gene should be blaOXA-51-like gene, CgMLST should be cgMLST.

Fixed as suggested

Round 2

Reviewer 1 Report

In the revised manuscript, Authors addressed only in part the major comments raised by reviewers and were not able to revised extensively their manuscript. In particular, they were not able to correctly address the following issues and to avoid misinterpretations throughout the manuscript:

1.       “Before answering the comments, we would like to make clear the goal of this review. It is dedicated not to genomic epidemiology of A. baumannii in general, but rather to providing definitions and the methods of assigning the isolates to several clonal lineages known as international clones of high risk (IC). Such definitions are usually made using cluster analysis, and sometimes are used in a controversial way. We would like to provide the readers, which could possibly be new to the field, with a comprehensive description of possible IC definitions and describe the procedure of assigning a particular isolate to known ICs.  Thus, the description of typing methods includes only the ones that can possibly be applied to this purpose, while other approaches not considered in the review could be suitable, for example, for outbreak investigation or other purposes. Genbank analysis was performed just for illustration purposes since this genomic set cannot be considered as representative for the whole population of A. baumannii in the world.” I disagree with the above comment done by Authors in addressing reviewer’s comment. In my opinion, genomic epidemiology is an useful tool to type Acinetobacter baumannii clinical isolates and assign them to clonal lineages.

2.       Table 1; The title is still confusing. It is my opinion that MALDI-TOF is an identification method, while it is not a typing method. This has been demonstrated in all references quoted by Authors in responmse to reviewer. Trilocus sequencebased typing (3-LST) is based on both DNA binding pattern and DNA sequence of amplified alleles. DNA allele sequence is mandatory to assign clonal lineages. MLST, blaOXA-51-like gene SLT and KL and OCL loci typing are all based on DNA sequencing of “complete” alleles. In case of KL and OCL loci typing, it is important to sequence the entire genomic loci and therefore genome sequencing is mandatory. In addition to cgMLST phylogeny and cluster analysis, whole genome sequencing is able to perform SNP (single Nucleotide Polimorphism)-based phylogeny. SNP-based phylogeny can be used to study isolates variations into the same lonal lineage as described by by Petazzoni et al. Microbiol Spectr. 2023 Mar 23:e0450522. doi: 10.1128/spectrum.04505-22. Moreover, SNP-based phylogeny has been extensively used to identify and assign A. baumannii isolates to clonal lineages as described in mSphere 2023-04-17 DOI: 10.1128/msphere.00098-23, FrontMicro 2019 doi: 10.3389/fmicb.2019.00930. However, Authors should be aware that cluster analysis and clonal lineage assignments in SNP-based phylogenies are performed through maximum likelihood analysis using an algorithm different from eBurst.

3.       Table 3 should be better described. Authors included in Table 3 data shown in original publications quoted as references 13, 28 and 46 of the manuscript. The above references should ber quoted. Also, it should be make clear that international clones are numbered according to the PasteurMLST and Diversi Lab typing REP PCR typing.

4.       Throughout the manuscript, eBURST analysis using Padsteur MLST scheme has been used to assign international clonal lineages. Because of this. the sentence that ICL3 is sometimes defined as CC3 or ST3, sometimes as CC124 or ST124 is confusing because ST124 is single locus variant of ST3.

My overall evaluation is that the review is not correctly performed and adds confusion to the field.

Minor editing of English language

Author Response

We would like to thank the reviewer for the time spend to analyze our manuscript. Our answers to the comments are provided below bold-faced.

In the revised manuscript, Authors addressed only in part the major comments raised by reviewers and were not able to revised extensively their manuscript. In particular, they were not able to correctly address the following issues and to avoid misinterpretations throughout the manuscript:

  1. “Before answering the comments, we would like to make clear the goal of this review. It is dedicated not to genomic epidemiology of A. baumanniiin general, but rather to providing definitions and the methods of assigning the isolates to several clonal lineages known as international clones of high risk (IC). Such definitions are usually made using cluster analysis, and sometimes are used in a controversial way. We would like to provide the readers, which could possibly be new to the field, with a comprehensive description of possible IC definitions and describe the procedure of assigning a particular isolate to known ICs.  Thus, the description of typing methods includes only the ones that can possibly be applied to this purpose, while other approaches not considered in the review could be suitable, for example, for outbreak investigation or other purposes. Genbank analysis was performed just for illustration purposes since this genomic set cannot be considered as representative for the whole population of A. baumannii in the world.”

I disagree with the above comment done by Authors in addressing reviewer’s comment. In my opinion, genomic epidemiology is an useful tool to type Acinetobacter baumannii clinical isolates and assign them to clonal lineages.

We think that the authors and the respectable reviewer have different definitions of genomic epidemiology in general and its applicability for particular studies in particular. However, this issue is not pertinent to description of suitable methods and approaches to IC assignments. We introduced changes into Table 1 and added new chapter 6 to remove the possible misunderstanding. 

  1. Table 1; The title is still confusing. It is my opinion that MALDI-TOF is an identification method, while it is not a typing method. This has been demonstrated in all references quoted by Authors in responmse to reviewer. Trilocus sequencebased typing (3-LST) is based on both DNA binding pattern and DNA sequence of amplified alleles. DNA allele sequence is mandatory to assign clonal lineages. MLST, blaOXA-51-like gene SLT and KL and OCL loci typing are all based on DNA sequencing of “complete” alleles. In case of KL and OCL loci typing, it is important to sequence the entire genomic loci and therefore genome sequencing is mandatory. In addition to cgMLST phylogeny and cluster analysis, whole genome sequencing is able to perform SNP (single Nucleotide Polimorphism)-based phylogeny. SNP-based phylogeny can be used to study isolates variations into the same lonal lineage as described by by Petazzoni et al. Microbiol Spectr. 2023 Mar 23:e0450522. doi: 10.1128/spectrum.04505-22. Moreover, SNP-based phylogeny has been extensively used to identify and assign A. baumannii isolates to clonal lineages as described in mSphere 2023-04-17 DOI: 10.1128/msphere.00098-23, FrontMicro 2019 doi: 10.3389/fmicb.2019.00930. However, Authors should be aware that cluster analysis and clonal lineage assignments in SNP-based phylogenies are performed through maximum likelihood analysis using an algorithm different from eBurst.

We removed MALDI-TOF from the table and changed the description of 3-LST and KL according to the comments provided. We also added SNP-based methods to the table.

  1. Table 3 should be better described. Authors included in Table 3 data shown in original publications quoted as references 13, 28 and 46 of the manuscript. The above references should ber quoted. Also, it should be make clear that international clones are numbered according to the PasteurMLST and Diversi Lab typing REP PCR typing.

We added the references and the statement regarding IC numbering near the table 3 in the text.

  1. Throughout the manuscript, eBURST analysis using Padsteur MLST scheme has been used to assign international clonal lineages. Because of this. the sentence that ICL3 is sometimes defined as CC3 or ST3, sometimes as CC124 or ST124 is confusing because ST124 is single locus variant of ST3.

By definition, clonal complex includes the central element and its single locus variants. Since both are determined using eBURST analysis for STs known at the time of such analysis, then the number of STs contained within particular CC could change from time to time when new STs become available (when new genomes or genome parts are uploaded to corresponding databases). The same could be true even for the central element itself since previous cluster analysis could become invalid.

We discussed this issue within novel chapter describing possible problems in IC identification, and we moved the statement for CC124 to this new chapter (chapter 6). In Table 3 we gave reference to CC3 only.

As for IC3, it was in fact attributed to CC3 by most researchers both recently and previously (e.g., 2022 - 10.1016/j.meegid.2021.105148,  10.1080/09603123.2021.1892036). At the same time, it was sometimes defined as CC124 (e.g.,

2023 - https://doi.org/10.1093/cid/ciad109;

Higgins 2022 -  https://www.peg-symposien.org/tl_files/symposien/Symposien_2022/Vortraege%20BHS%2022/Higgins%20PEG%202022_for%20pdf.pdf;

2019 -https://www.pmf.unizg.hr/images/50015941/IGB%201%20Crocmid%202019.pdf)

 We should state again, that our manuscript is not original article, but rather a review, which reflects the results obtained by other researchers, and thus can contain some contradictory information. For example, CC92Oxf for Oxford scheme is still used in epidemiological publications (e.g., 2022 - https://doi.org/10.3389, 2023 - https://doi.org/10.1186/s12941-023-00581-3), although it was shown to possibly be an artifact.

My overall evaluation is that the review is not correctly performed and adds confusion to the field.

We made changes into the manuscript to remove the possible confusion. We also added the chapter 6 describing common misconceptions and problems of IC determination to help the readers better understand this complex set of methods and definitions.

Reviewer 3 Report

Thanks for addressing all of my concerns. It is now suitable for publication.

Author Response

Thank you for your time and efforts in reviewing our manuscript.